# Cross-context qualitative synthesis of a mobile obstetric referral emergency system (MORES) in Ghana and Liberia

Camilla Bjelland[1]*, Joseph Sieka[2], Veronica Millicent Dzomeku[3], HaEun Lee[1],
Wahdae-Mai Harmon-Gray[2], Emmanuel K. Nakua[3], Jody R. Lori[1]

**1** University of Michigan, Ann Arbor, Michigan, United States of America, **2** University of Liberia College of Health Sciences, Monrovia, Liberia, **3** Kwame Nkrumah University of Science and Technology, Kumasi, Ghana

* cbjellan@umich.edu

pone.0350823

Pretoria, SOUTH AFRICA

**Peer Review History:** PLOS recognizes the
benefits of transparency in the peer review
process; therefore, we enable the publication
of all of the content of peer review and
author responses alongside final, published
articles. The editorial history of this article is
available here: https://doi.org/10.1371/journal.
pone.0350823

## Abstract

### Background

To address the delayed provision of obstetric emergency care in Ghana and Liberia, our team previously implemented a mobile obstetric emergency referral system (MORES) connecting rural health facilities (RHFs) and district hospitals through a South-South-North triangular collaboration. This secondary qualitative data analysis aimed to synthesize the perceived barriers and benefits associated with the adoption of MORES in Ghana and Liberia. To guide this cross-context synthesis, we utilized Rogers' Diffusion of Innovation Theory.

### Methods

The analysis included individual interviews conducted among 29 district hospital providers and 33 rural health facility (RHF) workers in Liberia. In Ghana, 11 midwives at a district hospital participated in a focus group discussion. The codes were developed utilizing an inductive process. Thematic analysis was utilized to compare adoption across countries through the Diffusion of Innovation Theory.

### Findings

Four themes were identified from the healthcare workers in Ghana and Liberia: compatibility, relative advantage, resource barriers, and limited implementation. According to healthcare workers, MORES was compatible with everyday workflow and had significant relative advantage, contributing positively to adoption. Healthcare workers were able to prioritize patient conditions upon presentation with increased referral information. Resource barriers and the limited implementation across districts restricted widespread adoption of MORES. Healthcare workers in both countries cited inconsistent access to ambulances as a barrier to complete obstetric referrals

**Data availability statement:** Public deposition of the data would breach compliance with the protocol approved by the research ethics board, specifically the participant consent. Upon reasonable request, the data may be available. Please email the open access data repository at the University of Michigan (deepblue@umich.edu) with data requests.

**Funding:** Research reported in this publication received funding support from the Bill and Melinda Gates Foundation (INV-023274, received by JL and JS) the USAID PEER Grant (project 9-518, received by VMD and JS) and the National Institute of Nursing Research of the National Institutes of Health (T32NR016914, received by CB). The funders had no role in study design, data collection and analysis, decision to publish, or preparation of the manuscript. https://www.gatesfoundation.org/ https://sites.nationalacademies.org/PGA/PEER/index.htm https://www.ninr.nih.gov/.

**Competing interests:** The authors have declared that no competing interests exist.

and in Liberia, the need for financial support for data and network connectivity was referenced. There were nuances by country, within the four themes. In Ghana, providers used MORES to facilitate follow-up on patients who did not complete referrals, contributing to MORES' relative advantage.

## Conclusions

MORES has the potential to reduce obstetric emergency referral delays in Ghana and Liberia. If the resource barriers and limited implementation of MORES are addressed, healthcare workers may continue to adopt and use the MORES system. Policy makers can address referral delays through the scale up of MORES, a compatible intervention with great relative advantage.

---

## Introduction

Smartphone use and mobile connectivity are rapidly expanding. By 2028, there will be a projected 7.9 billion smartphone mobile network subscriptions worldwide [1]. In 2024, 71% of the global population was estimated to own and use a smartphone [2]. Therefore, mobile health interventions (mHealth), using mobile network connectivity to improve communication and linkage to healthcare are becoming increasingly mainstream [3–5]. mHealth interventions have proved effective in low-and middle-income countries (LMICs) for improving maternal health outcomes [6,7]. mHealth improves health outcomes by linking users to essential sexual and reproductive services, providing personalized health information and promoting behavior change [8,9]. mHealth text messaging systems were effective in reminding women to have a skilled birth attendant and attend antenatal and postnatal care [8–11]. Further, mobile phone technology has been successfully utilized to connect women to obstetric emergency transportation services [12].

Strategic and innovative use of digital communication tools is instrumental in achieving the Sustainable Development Goals (SDGs) of reducing the global maternal mortality ratio (MMR) to less than 70 per 100,000 live births and ending preventable child and maternal mortality [13,14]. In 2023, sub-Saharan Africa accounted for 70% of maternal deaths [15]. The MMR in Ghana was estimated at 234 per 100,000 live births in 2023 [16]. Liberia, with one of the highest MMR in the world estimated at 628 in 2023, continues to struggle with reducing these deaths due to a decimated healthcare infrastructure following decades of conflict and the 2014−16 Ebola epidemic [16–18]. In both Ghana and Liberia, efficient escalation of care between health facilities during obstetric emergencies is needed to eliminate preventable deaths among women living in rural areas far from adequate healthcare centers [19–23]. Thus, the strategic implementation of digital communication tools in rural regions in both countries is vital.

To help tackle the issue of delayed provision of obstetric care contributing to maternal and child mortality in Liberia and Ghana, our team implemented the Mobile Obstetric Referral Emergency System (MORES) as part of a South-South-North or

"Triangular Cooperation" with the aim of generating actionable evidence that can be utilized by policymakers [24,25]. Triangular Cooperation involves a partnership between two global South countries, often with technical or specific research support from a Northern partner to collaborate, share knowledge and best practices, meet local needs for sustainable development, and expand research capacity [26,27]. The overall goal was to tackle a common problem identified in both countries, generate new knowledge, learn from each other, and strengthen research capacity.

Through Triangular Cooperation between researchers in Liberia, Ghana, and the United States, the MORES intervention was implemented on the WhatsApp platform to connect rural health facilities (RHFs) and district hospitals in Liberia and Ghana. The intervention was guided by the three delays of maternal mortality framework and specifically addressed the second and third delays, the delay of arrival at a health facility, and the delay in the provision of adequate care [25]. MORES provided a two-way communication system to facilitate prompt referral to the next level of care, allowing the facility to prepare for incoming patients and preventing delays upon arrival. The primary objective of this secondary qualitative analysis was to synthesize the perceived barriers and benefits associated with the adoption of MORES in and across contexts in Ghana and Liberia. The World Health Organization's (WHO) guidelines for digital health interventions emphasize the need for rigorous evaluation of digital health interventions to ensure the strengthening of health systems and improve health for all [28]. To guide this evaluation of implementation in Ghana and Liberia, we utilized the Diffusion of Innovation Theory developed by Rogers, which outlines key attributes that influence technology adoption [29]. This theoretical framework served as a valuable lens through which we evaluated and synthesized the adoption of MORES in Ghana and Liberia.

## Methods

### Study design, sample, and procedure

This study is a secondary data analysis of qualitative interviews one-year post-implementation of a Mobile Obstetric Referral Emergency System (MORES) in the Sene West and Sene East Districts of Ghana and Bong County, Liberia. MORES was implemented to facilitate efficient transfer of obstetric patients requiring emergency intervention. MORES connects rural health facilities (RHFs) to district hospitals, aiming to reduce prehospital and interfacility delays utilizing WhatsApp Messenger [30]. In Liberia, MORES was implemented among 20 RHFs and two district hospitals. In Ghana, MORES was implemented in nine RHFs in the Sene West District and seven RHFs in the Sene East District [31] and one district hospital, Kwame Danso. MORES' implementation was part of a larger study that additionally trained healthcare workers to triage obstetric patients according to the severity of their diagnosis [30]. The combination of the two interventions, obstetric triage training and the MORES intervention, ensured prompt identification, referral at the RHF, and a decrease in treatment delays at the district hospitals. At the district hospitals, one phone was placed in the emergency department. At RHFs, healthcare workers used their personal phones to complete referrals on WhatsApp and were provided cellular scratch cards for data [32]. More details on the implementation of MORES can be found in previously published articles from this project [30–33]. Additional information regarding the ethical, cultural, and scientific considerations specific to inclusivity in global research is included in the Supporting Information (S1 Checklist).

In Liberia, 10 research assistants (RAs) conducted the interviews in Bong County with healthcare providers one-year post-MORES implementation. Interviews were conducted in English. The RAs in Liberia were graduate students from the University of Liberia with experience conducting qualitative interviews. All healthcare workers from the 20 RHFs and 2 district hospitals were invited to participate in the interviews. The healthcare workers at the RHFs consisted of registered nurses and midwives while the healthcare workers at the district hospital consisted of registered nurses, midwives, physician assistants, and medical doctors [32]. In total, 62 individual interviews were conducted from March 20th to 24th, 2023, each lasting approximately 30 minutes. All 20 RHFs were represented in the interviews except one RHF, as healthcare workers from one RHF in the original training did not have access to a smartphone with WhatsApp capability and therefore did not complete referrals with WhatsApp [32].

In Ghana, an experienced RA proficient in Twi and English conducted the focus group discussions using a semi-structured interview guide. Midwives at Kwame Danso District Hospital who utilized the MORES system were invited to participate in a post-implementation focus group discussion. On October 16th, 2022, 11 midwives participated in the focus group lasting 1 hour and 50 minutes. All participants in both countries provided written informed consent.

The interview guides from both countries discussed the challenges and benefits of adopting the MORES intervention. Of the interview guide questions, six questions (~50%) were identical (S2 Interview Guide in S2 File). Questions differed as the interviews in Liberia included healthcare workers from both the referral hospital and the RHFs. The guide used in Liberia included specific questions asking how helpful feedback from the district hospital was to the RHF and the timing of the referral process. For efficiency, data were collected via focus group discussion in Ghana versus individual interviews in Liberia.

Physical data as well as electronic data from both countries were secured in a safe place. In Liberia, the interviews were audio-recorded and stored in a locked container for travel. Transcripts were transcribed verbatim and uploaded to a secure Dropbox. Similarly, in Ghana, the focus group was audio recorded, transcribed, and stored securely on a password-protected laptop. The password was known only to the original research team members.

## Analysis

An inductive process was utilized to develop the code list for the transcripts from Liberia by reading through initial transcripts and letting the data tell the story [34]. The code list was discussed by three team members (JL,HL,CB) finalized and applied to all individual transcripts in Liberia. All transcripts were imported into and coded in Dedoose (Version (9.2.22) [35]. All code frequencies were tabulated. Any codes that appeared in fewer than 7/62 transcripts (11%) were merged or dropped. All researchers agreed on the final codes and themes.

The process used to analyze the data from Liberia was replicated with the data from Ghana. Codes were created inductively and then applied to the focus group. All analysis of the Ghanaian focus group was completed in Word (Version 16.95.4). In the Ghanaian focus group transcript, the number of times each code was mentioned was counted, but it could not be determined how many different individuals made those references. Therefore, when participants were asked follow-up questions or if it was unclear whether a new person was speaking codes were counted conservatively. Codes referenced less than two times were merged or dropped.

## Theoretical framework

Rogers' Diffusion of Innovation theory was tested deductively in the context of MORES adoption to deeply understand the benefits and barriers of adoption versus non-adoption. To test the theory, all codes from both countries were considered if they mapped onto Rogers' Diffusion of Innovation Theory and compared across contexts [29]. In this way, Rogers' Diffusion of Innovation Theory was empirically examined [36]. When codes did not map onto Rogers' Diffusion of Innovation Theory, unique themes were reported. The Diffusion of Innovation was considered a generic proposition or a hypothesis about how the world works [37]. Rogers asserts the following five attributes, as perceived by members of a social system, determine its rate of adoption: (1) relative advantage, (2) compatibility, (3) complexity, (4) trialability, and (5) observability [29]. The innovation of interest is the adoption of MORES. Table 1 defines each Diffusion of Innovation attribute that was deductively tested according to the transcripts detailing satisfaction with utilizing MORES.

## Statement of reflexivity

An audit trail detailing personal, theoretical, and analytic memos was kept by the first author and discussed with members of the research team to ensure trustworthiness [34,38]. Participant checking was not possible due to the secondary nature of the data. The authors identify with critical constructivism, acknowledging the power dynamics that shape the socially constructed world, specifically the material world [39]. This study was a collaboration between researchers in the United States, Ghana, and Liberia. They worked together to ensure the results were trustworthy and accurately reflected the

**Table 1. Rogers' Diffusion of Innovation Theory Applied to MORES Adoption.**

| Rogers Concept | Application to MORES Innovation |
| --- | --- |
| Relative Advantage | Perceived advantages of using MORES for emergency referrals |
| Compatibility | Perceived consistency of the MORES intervention with the existing workflow and need |
| Complexity | Perceived difficulty of completing MORES emergency referrals |
| Trialability | The extent to which MORES could be experimented with before adoption |
| Observability | The extent to which healthcare workers are influenced to use MORES through exposure |

experiences of the healthcare workers in Ghana and Liberia. Further, the collaborative nature of this South-South-North research team provided multiple cultural and disciplinary perspectives during analysis, helping to challenge assumptions and ensure that interpretations were grounded in the local contexts of Ghana and Liberia.

### Ethical considerations

All participants were well informed about the research project and signed a consent form before beginning the interviews. Data were deidentified prior to analysis. The institutional review board at the University of Michigan approved this secondary analysis (ID: HUM00268431). The original study in Ghana was approved by the Ghana Health Services Review Committee (GHS-ERC: 004/03/21). In Liberia, both the University of Liberia (IRB00013730) and the University of Michigan (HUM0019059) approved the original study. The results of this study are reported according to COREQ (consolidated criteria for reporting qualitative research) guidelines [40].

## Results

Demographic questionnaires were not administered to the 11 midwives who participated in the focus group discussion in Ghana. In Liberia, the mean age of the healthcare workers was 39.2, and the majority of participants were midwives (n = 38, 61%). The time in practice ranged from less than three years to 20 years. Overall, 33 of the participants worked at RHFs, and 29 of the participants worked at district hospitals. Table 2 details the healthcare worker demographics from Liberia.

### Themes

All themes that were congruent or divergent from the Diffusion of Innovation Theory are reported below in Table 3. Complexity, Trialability, and Observability were not themes present in the data. Completing referrals with the MORES platform was not reported as complex. However, healthcare workers in Ghana and Liberia discussed their respective needs to effectively utilize the WhatsApp platform. Two healthcare workers from Liberia and two healthcare workers from Ghana were utilizing MORES, though they were not initially trained. This potentially indicates trialability, though this was not explicitly stated by participants. Further, there was no specific discussion of being influenced to use MORES because of peer influence (observability) although this is very likely the case.

### Relative advantage

The relative advantage of the MORES intervention was apparent in both countries. In Liberia, MORES was helpful to prepare for a referral, as asserted among rural healthcare workers and district healthcare workers, and prevalent in 42/62

**Table 2. Healthcare worker demographics of those who participated in individual interviews in Liberia.**

| Total | N = 62 |
|---|---|
| Age (years), mean (SD) | 39.2 (7.22) |
| Role, n (%) | |
| Registered nurse | 20 (32) |
| Midwife | 38 (61) |
| Physician Assistant | 3(5) |
| Medical doctor | 1 (2) |
| Times in practice (years), mean (SD) | 7.8 (3.61) |

transcripts (68%). Rural healthcare workers commented on how prepared the hospital was for the patient's arrival. Further, a healthcare worker at a district hospital remarked,

"I'm at the referral hospital, and if a patient comes without you knowing, it is challenging. You'll be running around to get things in order, especially if it's a bleeding case. But with this [MORES intervention], as soon as we receive the message, we get up and place one person in the ER awaiting the ambulance. As soon as the ambulance arrives, we start our emergency care, unlike those places without the program. For those clinics on the WhatsApp platform, it is very easy to cater to their patients."

In Ghana, the code *helpful to prepare* was referenced 6 times. One healthcare worker remarked,

"It saves time in the sense that as soon as the client is referred, you will see the information, so you will prepare. You won't wait till the client comes. If the woman is in labor, for instance, you won't run to pick oxygen because you have already prepared everything, and this will save the client's life."

Healthcare workers in both Ghana and Liberia specifically referenced reducing preventable maternal and child mortality (Liberia: 33/62, 53%, Ghana: 5 references) through MORES utilization. In Liberia, a healthcare worker remarked,

"It has helped me to save lives maybe that were to be lost, especially in pregnant women cases. If someone has a ruptured ectopic, before the person comes, we already know the person's condition. The OR team will be informed, and everybody will be set, and the person will receive the care in time."

A Ghanaian healthcare worker shared, "It has also come to reduce some of the preventable mortality and morbidity."
A unique perspective, shared by Ghanaian healthcare workers was that the MORES intervention allowed healthcare workers to follow up on patients who did not complete obstetric referrals to Kwame Danso District Hospital (7 references). Healthcare workers at the district hospital contacted the RHF to report on patients not arriving at the referral hospital,

"They [patients] will go and sit in the house and delay and sometimes when they come what you don't want to happen has already happened. But because right now they put it on this [WhatsApp] page, we will tell them [rural healthcare workers] that the client that you referred has not come. Because they are closer to them, they can go to their house for them so that we know about any complications that will lead to mortality."

Further, healthcare workers in Ghana felt that the MORES intervention was a better system providing more detailed information and ensuring confidentiality (3 references). For example, one healthcare worker said, "Okay, for the WhatsApp

**Table 3. Themes and codes from Ghana and Liberia testing the Diffusion of Innovation Theory.**

| Diffusion of Innovation and Inductive Themes | Liberia-Specific Findings | Ghana-Specific Findings | Ghana and Liberia Similarities |
|---|---|---|---|
| Relative advantage | **Feedback:** communication increased knowledge of rural healthcare workers: 11/62 | **Follow through:** healthcare workers at the hospital could communicate with rural healthcare workers to check in on women who did not complete/ delayed their referral: 7 references<br>**Better system:** MORES is more detailed and confidential than the previous system: 3 references | **Helpful to prepare:** WhatsApp communication helped healthcare workers to prepare to receive patients at district hospitals<br>Liberia: 42/62<br>Ghana: 6 references<br>**Reduce maternal and child mortality:** innovation decreases maternal and child mortality and addresses maternal complications<br>Liberia: 33/62<br>Ghana: 5 references |
| Compatibility | **Bi-directional feedback:** communication between facilities met the needs of healthcare workers<br>Acceptable timing, same day or next day feedback: 21/62<br>Difficult to receive feedback: 7/62 | **Competing responsibilities:** difficult to find time to answer WhatsApp messages: 6 references | **Collaboration:** communication between facilities improves relationships and patient care<br>Liberia: 11/62<br>Ghana: 2 references<br>**Timely:** obstetric emergency patients waited less time to receive care at the hospital<br>Decreased referral/waiting time:<br>Liberia: 29/62<br>Prioritize who needed care first:<br>Ghana: 12 references |
| Resource barriers | **Lack of resources:** needs that prevent the use of WhatsApp and caring for patients<br>Network connectivity: 20/62<br>Need healthcare materials (drugs, fuel): 15/62<br>Need credits/data scratch cards: 31/62 | Not reported | **Access to phone:**<br>healthcare workers need increased access to phones<br>Liberia:14/62<br>Ghana: 3 references<br>**Transportation difficulties:**<br>need district ambulance; hardship of hiring an individual car<br>Liberia: 26/62<br>Ghana: 3 references<br>**Need motivation:** financial motivation; emotional support<br>Liberia:15/62<br>Ghana: 3 references |
| Limited implementation | Not reported | Not reported | **Desire more training:** training for those not initially trained; reinforcement training<br>Liberia: 14/62<br>Ghana: 3 references<br>**Desire expansion:** scale up of intervention to extend to the surrounding area<br>Liberia: 19/62<br>Ghana: 3 references |

program, one thing I have realized about it is that it has been more detailed than the previous way we used to refer clients and more confidential". Before the MORES intervention, women would arrive with or without a referral letter, which provided very little information for midwives regarding the patient's diagnosis (4 references).

Rural healthcare workers in Liberia stated that feedback received on referred patients increased their knowledge (11/33 rural healthcare workers, 33%, 11/62 all healthcare workers, 18%). For example, a healthcare worker from a RHF said, "It is helpful because it makes you know whether my suspicion was right or not. And if I see the condition, I will know how to address it." This newfound knowledge is expected to change the way healthcare workers practice in the future.

## Compatibility

Healthcare workers in Ghana and Liberia discussed how the MORES intervention was easily integrated into their existing workflows, making the obstetric referral process timelier. Obstetric emergency patients waited less time to receive care at the hospital. In Ghana, healthcare workers emphasized their ability to prioritize who needed care first with their increased knowledge of the patient's condition (12 references). One healthcare worker commented, "It makes us know more details about the client and whom to treat urgently. Although all cases are important some need much more attention." In Liberia, timeliness was emphasized differently; healthcare workers mentioned decreased referral/waiting times for patients to receive care (29/62, 47%). A healthcare worker at a RHF mentioned, "It is useful for caring for pregnant women because it shortens the time pregnant women used to stay before receiving care."

Overall, the bi-directional feedback loop, communication between the district and RHFs, was perceived as satisfactory by rural healthcare providers in Liberia. Most rural healthcare workers (21/33, 54%) and 21/62 of all healthcare workers, (34%) mentioned being pleased with the feedback they received, receiving updates within a day or two on patient conditions. A healthcare worker from a RHF reported that getting feedback "takes less than five minutes". However, 7/33 rural health workers (21%, 7/62 all healthcare workers,11%) mentioned that receiving feedback was a difficulty. A different healthcare worker from a RHF remarked, "Some days we have to be asking in the group chat; like 3-4 days before even getting feedback".

Healthcare workers in Ghana described their competing responsibilities as a barrier to finding time to answer MORES messages (6 references). For example, one healthcare worker, when discussing her shift, shared, "The phone is not with you. So, when they refer a case, before we get to the phone, you see the client is here. That's the only downside I can say. We don't have enough time to check the phone."

## Resource barriers

In Liberia, healthcare workers reported needing credits/data scratch cards (31/62, 50%), and obstetric emergency materials such as drugs, fluids, and fuel for transportation (15/62, 24%). A healthcare worker at a district hospital spoke to this need, "We need materials like fluids. Sometimes relatives come without materials which are essentially used in the obstetric ward." A healthcare worker from a RHF commented on the inability to utilize MORES without data, "I can use the program once there's data on my phone. The only time I don't use it is when [there is] no data." Further, lack of network connectivity inhibited prompt communication with MORES (20/62, 32%). When asked what could make the program better, a healthcare worker replied, "Work along with GSM (cellular network) companies to improve coverage in areas where there is no network." This healthcare worker previously mentioned, "Some of our colleagues have to walk for over 10 minutes [to find network connection] before communicating."

Accessing designated intervention phones at the district hospitals in both Liberia and Ghana was difficult (Liberia: 14/62, 23%, Ghana: 3 references). In Ghana, a healthcare worker commented, "I think at least everybody should have access to the phone. So that we can be able to check for referral messages." In Liberia, there was a similar desire for increased accessibility to phones,

> "We need another phone for the labor ward because the WhatsApp phone is in the ER…Also, people who have smartphones could be added to the WhatsApp group they can see the messages just in the case that someone is not around the phone."

Transportation difficulties for transferring patients in both Ghana and Liberia were apparent. Once the referral was initiated it was often difficult to complete in Liberia (26/62, 42%). A healthcare worker from a RHF commented, "But for the ambulance system, sometimes it was hard to get the ambulance. You'll have to tell the patients to transport themselves." Transportation difficulties were referenced 3 times in Ghana. A healthcare worker reported, "Because some of them

[patients], do not have money for transportation. Somebody was referred, she was 42 weeks, but she didn't have money to come."

### Limited implementation

Further, there was an apparent desire for more training in both countries. The training was referenced 3 times in Ghana and 14/62 transcripts (23%) in Liberia. A Ghanaian healthcare worker commented, "If we can also have the training again, I will be happy because some of our colleagues weren't fortunate enough to get trained." Healthcare workers from both countries additionally wanted MORES to expand into more settings and scale up to more healthcare workers (Liberia: 19/62, 31%; Ghana: 3 references) "If we introduce it nationwide, it will help because sometimes we are the referral point. We also refer to other hospitals, so letting other hospitals know who will receive the patient would also be good," said a Ghanaian healthcare worker. In Liberia, a healthcare worker at a RHF reflected, "What I think to make this program better is to include other facilities, it will also help. Because some of our friends are doing referrals and they want to know updates about that patient."

## Discussion

### Theory evaluation

In the context of MORES, the compatibility of WhatsApp and the relative advantage of utilizing MORES were prominent themes. However, Roger's constructs of complexity, trialability, and observability were not themes that emerged inductively from the transcripts. Rogers highlights that while all five qualities of an innovation are meaningful for adoption, relative advantages and compatibility are of paramount importance [29]. Additionally, two themes that emerged inductively, resource barriers and limited implementation, did not map onto the Rogers Theory of Innovation. While Rogers points out that the initial cost of innovation may affect its rate of adoption, in this case, it was not an initial cost but resource barriers for maintaining the innovation (data scratch cards) and limited implementation that hindered widespread adoption [29]. These findings are consistent with a systematic review on the diffusion of innovation in the private sector (industry and services) in LMICs, which found economic characteristics such as the level of economic development and infrastructure as potentially preventing innovation diffusion [41]. Thus, the following explanation of the barriers and benefits of adopting the innovation MORES is presented (Fig 1). In Ghana and Liberia, relative advantage and compatibility of the innovation contributed positively to the adoption of MORES, while resource barriers and limited implementation hindered the adoption of completing obstetric emergency referrals with MORES.

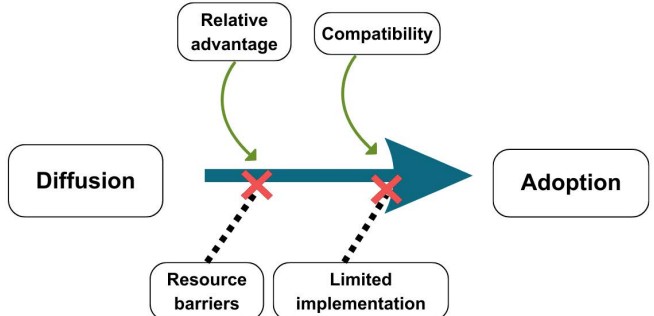

**Fig 1. Barriers and benefits to adopting MORES in Ghana and Liberia.**

By deductively testing the Diffusion of Innovation Theory, the results present a refinement of the theory [36]. The refinement of the Diffusion of Innovation Theory presented may be transferable to evaluating the adoption of an innovation in similar contexts.

The WHO's guidelines on digital interventions for health system strengthening regarding provider-to-provider communication suggest that healthcare workers appreciate the opportunities to communicate with each other, and lower-level healthcare workers appreciate being able to ask for advice from higher-level workers [42]. This notion is consistent with the findings of this study. Healthcare workers in Liberia (11/62, 18%) reported that the feedback they received on their referral patients, made them aware of their correct diagnosis or misdiagnosis and was helpful for their learning. Further, evidence on the feasibility of digital health interventions is related to network connectivity, access to electricity, usability of the device, and sustaining training [43].

Resource barriers such as transportation and network connectivity persist and influence the second delay, the delay in patients reaching care in Ghana and Liberia [25]. Roughly a third (20/62, 32%) of healthcare providers in Liberia mentioned network connectivity as a barrier to utilizing MORES to facilitate referrals. This was previously recorded as a perceived fear of healthcare workers in Ghana and a barrier to using MORES in Liberia [31,33]. Healthcare providers in both Liberia and Ghana discussed difficulties with transportation. In a previous study of WhatsApp implementation in the Greater Accra region of Ghana, the majority of referred patients (80.2%) arrived by public transportation despite the 2004 Ghanaian launch of a national ambulance service and the median referral-to-arrival time was significantly shorter for those who travelled by ambulance [44,45]. Our team's research on MORES quantitative outcomes in Liberia highlighted only 12.9% of women arrived at the hospital within 2 hours or less [30]. The timeframe of 2 hours is *The Lancet's* definition of geographically accessible healthcare as it is a rough estimate from the onset of bleeding to death in post-partum hemorrhage without intervention [46,47]. This is similar to the study in the Greater Accra region of Ghana, which found only 23.5% of patients arrived within 2 hours when WhatsApp was used to facilitate their referral [45]. Attention needs to be paid to the challenges of the referral transportation system in obstetric emergencies, mirroring the findings of Dzomeku and colleagues' qualitative study from Ghana [48].

Despite the resource barriers, MORES addressed the delay in provision of care, a concept from the WHO Health System Challenge framework to guide digital health interventions, and the third delay in preventable maternal death [25,42]. This finding aligns with previous research from the MORES study in Liberia, which highlights how MORES improves the provision of timely care [30]. There was an association between MORES and a significant increase in timely cesarean sections and a reduction of newborns with poor respiratory effort, muscle tone, or heart rate [30]. Similarly, in Uganda, a referral intervention through phone calls reduced the delay of patients receiving care and was protective against adverse maternal-fetal outcomes [49]. This study reinforces a policy brief recommendation that the Liberian Ministry of Health should invest in MORES, scaling it up in Bong County and expanding implementation to other counties [53].

Further, the following Health System Challenges constructs were subsequently addressed through MORES implementation: continuity of care, planning and coordination, and access to information [42]. Ineffective communication between health facilities and hospitals, a contextual cause of maternal death in Bong County, identified by a recent verbal autopsy analysis, was addressed through MORES [50]. MORES was effective at strengthening healthcare communication between RHF and district hospitals similar to other mHealth referral interventions. A systematic review of obstetric emergency communication and feedback between health facilities in sub-Saharan Africa found that referral communication using mobile phones and referral letters/notes resulted in increased communication [51]. The WhatsApp platform has previously been utilized to improve communication during obstetric emergency referrals in Ghana [45].

This secondary qualitative synthesis highlights the unique insights generated through this South-South-North collaboration. In Ghana, providers used MORES to facilitate follow-up on patients who did not complete referrals with rural healthcare workers. Previous qualitative research from the Sene East and West Districts of Ghana identified the need for patients to comply with referral directives [48]. MORES addressed this need. Lessons learned from Ghana on the

close tracking of patients not completing the referral process were shared with Liberia. Learnings from one Global Southern country are often more relevant to another country in the Global South than to a Global Northern country [27]. The researchers in the South-South-North collaboration appreciated the opportunity to leverage similarities in healthcare systems and address patient needs in both countries in the Global South. This collaboration has led to additional joint maternal health projects between the researchers from Ghana and Liberia.

## Strengths and limitations

This study had several limitations. First, a weakness of this study is the use of two different data collection approaches. In Liberia, data were collected in 10-minute individual interviews from both healthcare workers at rural and district health facilities while in Ghana, focus group discussion was used with midwives at only the district hospital. There was unequal representation from each country. In Liberia, 62 healthcare workers were interviewed and 11 healthcare workers participated in the focus group Ghana. Therefore, the findings of the study should be interpreted as a cross-context synthesis rather than a comparison across countries.

While the qualitative data differed in structure from each country, a strength of the study is the inclusion of healthcare workers' perspectives with similar systems in Ghana and Liberia. Ghana and Liberia both employ a referral process between basic emergency obstetric and newborn care (BEmONC) facilities and comprehensive emergency obstetric and newborn care facilities (CEmONC) when an escalation of care is needed for blood transfusion and or cesarean section [19,47,52]. A notable strength of this study is the applied cross-country learnings.

## Conclusion

This study found MORES has the potential to reduce obstetric emergency delays in care, and healthcare workers in Ghana and Liberia felt it had a great relative advantage over current systems and was compatible with their workflow. To reduce preventable child and maternal mortality and achieve the SDGs, governments and private sector partners in Ghana and Liberia must work together to scale up MORES by: (1) negotiating with telecommunications companies to improve network coverage in rural areas, (2) providing consistent funding for data credits, and (3) strengthening the ambulance referral system to ensure that once a referral is initiated, transportation is reliably available.

## Supporting information

**S1 Checklist. Checklist.**
(DOCX)

**S2 File. Interview Guide.**
(DOCX)

## Acknowledgments

The authors would like to thank the healthcare workers at the district hospitals and rural health facilities in Ghana and Liberia for their commitment, dedication, and hard work caring well for their patients, as well as the students and research assistants who collected the data.

## Author contributions

**Conceptualization:** Camilla Bjelland, Joseph Sieka, Veronica Millicent Dzomeku, HaEun Lee, Jody R. Lori.

**Data curation:** Joseph Sieka, Veronica Millicent Dzomeku, Wahdae-Mai Harmon-Gray, Emmanuel K. Nakua, Jody R. Lori.

**Formal analysis:** Camilla Bjelland.

**Funding acquisition:** Joseph Sieka, Veronica Millicent Dzomeku, Jody R. Lori.

**Investigation:** Joseph Sieka, Veronica Millicent Dzomeku.

**Methodology:** Veronica Millicent Dzomeku, HaEun Lee, Jody R. Lori.

**Project administration:** Joseph Sieka, Wahdae-Mai Harmon-Gray, Emmanuel K. Nakua, Jody R. Lori.

**Supervision:** HaEun Lee.

**Visualization:** Camilla Bjelland.

**Writing – original draft:** Camilla Bjelland.

**Writing – review & editing:** Joseph Sieka, Veronica Millicent Dzomeku, HaEun Lee, Wahdae-Mai Harmon-Gray, Emmanuel K. Nakua, Jody R. Lori.

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
