## [Decision Letter · Decision Letter 0]

6 Jan 2026

PONE-D-25-52452Comparative analysis of a mobile obstetric emergency referral system (MORES) in Ghana and LiberiaPLOS One

Dear Dr. Bjelland,

Thank you for submitting your manuscript to PLOS ONE. After careful consideration, we feel that it has merit but does not fully meet PLOS ONE’s publication criteria as it currently stands. Therefore, we invite you to submit a revised version of the manuscript that addresses the points raised during the review process.

A letter that responds to each point raised by the academic editor and reviewer(s). You should upload this letter as a separate file labeled ’Response to Reviewers’.A marked-up copy of your manuscript that highlights changes made to the original version. You should upload this as a separate file labeled ’Revised Manuscript with Track Changes’.An unmarked version of your revised paper without tracked changes. You should upload this as a separate file labeled ’Manuscript’.

We look forward to receiving your revised manuscript.

Kind regards,

Jianhong Zhou

Staff Editor

PLOS One

Journal Requirements:

1.Please ensure that your manuscript meets PLOS ONE’s style requirements, including those for file naming. The PLOS ONE style templates can be found at

3. Please expand the acronym “USAID” (as indicated in your financial disclosure) so that it states the name of your funders in full.

Additional Editor Comments:

The manuscript has been evaluated by two reviewers, and their comments are available below.

The reviewers have raised a number of major concerns. They feel the manuscript should clearly justify the substantial difference in sample size and data collection methods.

Reviewers’ comments:

Reviewer’s Responses to Questions

**Comments to the Author**

1. Is the manuscript technically sound, and do the data support the conclusions?

Reviewer #1: Partly

Reviewer #2: Yes

2. Has the statistical analysis been performed appropriately and rigorously? 

Reviewer #1: N/A

Reviewer #2: N/A

3. Have the authors made all data underlying the findings in their manuscript fully available?

Reviewer #1: No

Reviewer #2: Yes

4. Is the manuscript presented in an intelligible fashion and written in standard English?

Reviewer #1: Yes

Reviewer #2: Yes

5. Review Comments to the Author

Reviewer #1: This manuscript addresses an important and policy-relevant topic: the use of mHealth to strengthen obstetric emergency referral systems in low-resource settings. The focus on Ghana and Liberia and the description of a South–South–North collaboration are valuable contributions. The manuscript is generally well written and the qualitative findings particularly from Liberia provide useful insight into healthcare workers’ experiences with MORES.

That said, several issues related to study design, analytic framing, and reporting clarity should be addressed to strengthen the manuscript and ensure that conclusions are well aligned with the data.

1. While the Methods section later clarifies that this study is a secondary analysis of previously collected qualitative data, the abstract reads as though primary data collection was conducted specifically for the current study (e.g., “interviews were conducted,” “midwives participated in a focus group”). Given the methodological implications of secondary qualitative analysis, this distinction should be made explicit in the abstract for transparency and accurate interpretation.

2. The manuscript is framed as a comparative analysis; however, the datasets differ substantially between countries. Liberia’s data comprise 62 individual interviews across both rural health facilities and district hospitals, while Ghana’s data derive from a single focus group discussion with 11 midwives at one district hospital. These differences limit the extent to which findings can be directly compared across countries. The authors are encouraged to more clearly justify the comparative framing, temper comparative claims, or reframe the analysis as a cross-context qualitative synthesis.

3. The manuscript reports frequency counts and percentages for Liberia while using raw reference counts for Ghana. Given the substantial difference in sample size and data collection methods, numerical comparisons across countries should be avoided or more clearly contextualized to prevent overinterpretation.

4. The rationale for restricting public access to transcripts due to ethical considerations is understandable; however, the Data Availability Statement would benefit from clearer alignment with PLOS ONE’s policy language. Clarifying whether de-identified excerpts or controlled access might be possible would strengthen transparency.

5. Some global statistics in the introduction (e.g., smartphone subscriptions) should be carefully checked for accuracy. It probably should be 7.4 billion and not million as stated.

Overall, this is a promising manuscript with clear practical relevance. Addressing the points above would substantially strengthen its methodological transparency, analytic coherence and contribution to the literature.

Reviewer #2: The study aimed at determining benefits and barriers in using a mobile obstetric emergency referral system (MORES) from the perspective of healthcare workers in Liberia and Ghana, two countries with significantly high maternal mortality. While the target population was the same, the methods used were different: individual interviews in Liberia and a focus group discussion in Ghana. The findings revealed that the use of MORES facilitated the continuity of care for pregnant mothers and expedited referrals to improve timely intervention. Limitations in resources, rather than in technology adoption, were cited as the main barriers. These results are consistent with those of related studies looking into the role of technology in mitigating maternal deaths and validates previous recommendations made.

The authors should avoid doing citations in the Methodology section and never in the Conclusions section.

The Results seem to point that the Diffusion of Innovation Theory may not be the most appropriate theoretical framework as three of its categories are not significantly touched on and that there are themes captured but are not in the theory. Perhaps this should be explained better in the Discussion section rather than just mentioned as a matter of fact.

The differences in methods of data collection between Liberia and Ghana are mentioned as part of the study limitations but how such differences impact the findings need to be further elaborated.

Still in the Strengths and Limitations Subsection, the two paragraphs on limitations seem to be contradicting with each other. The first paragraph says that the results should be interpreted carefully yet the second paragraph is saying that the findings can be generalized. These should be reviewed.

Overall, this paper can be accepted for publication as is, but can still be improved with some very minor revisions.

6. PLOS authors have the option to publish the peer review history of their article (what does this mean?). If published, this will include your full peer review and any attached files.

Reviewer #1: **Yes:**Final Zimkhitha Juqu

Reviewer #2: No

---

## [Author Response · Author response to Decision Letter 1]

10 Apr 2026

Editor comment:

The reviewers have raised a number of major concerns. They feel the manuscript should clearly justify the substantial difference in sample size and data collection methods.

Response:

Thank you for your response. We clarified that our study is a “cross-context synthesis” rather than a comparative analysis due to the difference in sample size (title, abstract line 19, and strengths and limitations 716). We believe this will help temper overinterpretation of the findings. We also detailed this limitation in the Strengths and Limitations section.

Reviewer #1 comments:

While the Methods section later clarifies that this study is a secondary analysis of previously collected qualitative data, the abstract reads as though primary data collection was conducted specifically for the current study (e.g., “interviews were conducted,” “midwives participated in a focus group”). Given the methodological implications of secondary qualitative analysis, this distinction should be made explicit in the abstract for transparency and accurate interpretation

Response:

Thank you for pointing out our lack of clarity about the nature of the study in the abstract. The abstract on page 1-2 has been corrected to clearly reflect the study design.

Line 14-17 now reads, “To address the delayed provision of obstetric emergency care in Ghana and Liberia, our team previously implemented a mobile obstetric emergency referral system (MORES) connecting rural health facilities (RHFs) and district hospitals through a South-South-North triangular collaboration. This secondary qualitative data analysis aimed to synthesize the perceived barriers and benefits associated with the adoption of MORES in Ghana and Liberia.”

Comment:

The manuscript is framed as a comparative analysis; however, the datasets differ substantially between countries. Liberia’s data comprise 62 individual interviews across both rural health facilities and district hospitals, while Ghana’s data derive from a single focus group discussion with 11 midwives at one district hospital. These differences limit the extent to which findings can be directly compared across countries. The authors are encouraged to more clearly justify the comparative framing, temper comparative claims, or reframe the analysis as a cross-context qualitative synthesis.

Response:

Thank you for your thoughtful feedback regarding the framing of the paper. We have adjusted the wording to frame the paper as a cross-context synthesis. The title was changed to: “Cross-context qualitative synthesis of a mobile obstetric referral emergency system (MORES) in Ghana and Liberia” Line 19 reads, “To guide this cross-context synthesis, we utilized Rogers’ Diffusion of Innovation Theory.” Further in the Strengths and Limitations section, a distinction is made to list the different data sources as a limitation.

Line 715-716 now reads, “Therefore, the findings of the study should be interpreted as a cross-context synthesis rather than a comparison across countries.”

Comment:

The manuscript reports frequency counts and percentages for Liberia while using raw reference counts for Ghana. Given the substantial difference in sample size and data collection methods, numerical comparisons across countries should be avoided or more clearly contextualized to prevent overinterpretation.

Response:

We agree it is important to contextualize our methods to prevent overinterpretation. We believe the frequency counts, percentages and reference counts increase transparency in our conclusions since the data set sample sizes significantly differed between the countries. The difference in sample size in addressed in our Strengths and Limitations section. We added more explanation as to why the sample sizes were different in the Methods Line 312-313 “For efficiency, data were collected via focus group discussion in Ghana versus individual interviews in Liberia.”

Comment:

The rationale for restricting public access to transcripts due to ethical considerations is understandable; however, the Data Availability Statement would benefit from clearer alignment with PLOS ONE’s policy language. Clarifying whether de-identified excerpts or controlled access might be possible would strengthen transparency.

Response:

Thank you for your comment. We have revised our data availability statement to further clarify. “The research data is not available online due to the wording in the participant consent. Upon reasonable request, the data may be available. Please email the corresponding author and the open access data repository at the University of Michigan (deepblue@umich.edu) with such requests.”

Comment:

Some global statistics in the introduction (e.g., smartphone subscriptions) should be carefully checked for accuracy. It probably should be 7.4 billion and not million as stated.

Response:

Thank you for catching this typo. We updated this statistic further. Line 54-55: “By 2028, there will be a projected 7.9 billion smartphone mobile network subscriptions worldwide.”

Reviewer #2 comment:

The authors should avoid doing citations in the Methodology section and never in the Conclusions section.

Response:

Thank you for your comment. The citation was removed from the conclusion section. We feel the citations in the methodology section establishes credibility. Since this is a secondary analysis, we acknowledge the previous work our team has done through citations.

Comment:

The Results seem to point that the Diffusion of Innovation Theory may not be the most appropriate theoretical framework as three of its categories are not significantly touched on and that there are themes captured but are not in the theory. Perhaps this should be explained better in the Discussion section rather than just mentioned as a matter of fact.

Response:

Thank you for your insightful comment. Through inductive analysis of the codes in the transcripts, the three constructs complexity, trialability, and observability were not discussed. We cannot be certain that those constructs did not influence adoption, only that they were not brought up in the interviews. Roger’s Diffusion of Innovation was extremely helpful for our understanding of the ways constructs hinder or encourage adoption. We have added further clarity regarding your point. Lines 560-564 now reads: “In the context of MORES, the compatibility of WhatsApp and the relative advantage of utilizing MORES were prominent themes. However, Roger’s constructs of complexity, trialability, and observability were not themes that emerged inductively from the transcripts. Rogers highlights that while all five qualities of an innovation are meaningful for adoption, relative advantages and compatibility are of paramount importance [29].”

Comment:

The differences in methods of data collection between Liberia and Ghana are mentioned as part of the study limitations but how such differences impact the findings need to be further elaborated.

Response:

Thank you for this comment. Reviewer 1 also highlighted this point and we agree a reframing of the study to avoid overinterpretation is necessary. We changed the language around this study being a comparison to being a “a cross-context synthesis” (title, abstract line 19, and strengths and limitations 716).

Comment:

Still in the Strengths and Limitations Subsection, the two paragraphs on limitations seem to be contradicting with each other. The first paragraph says that the results should be interpreted carefully yet the second paragraph is saying that the findings can be generalized. These should be reviewed.

Response:

We appreciate you highlighting our contradiction. The language in the Strengths and Limitations section has been edited for clarity. Lines 642-655, end of paragraph 1 and paragraph 2 in this section read: “There was unequal representation from each country. In Liberia, 62 healthcare workers were interviewed and 11 healthcare workers participated in the focus group Ghana. Therefore, the findings of the study should be interpreted as a cross-context synthesis rather than a comparison across countries.

While the qualitative data differed in structure from each country, a strength of the study is the inclusion of healthcare workers’ perspectives with similar systems in Ghana and Liberia. Ghana and Liberia both employ a referral process between basic emergency obstetric and newborn care (BEmONC) facilities and comprehensive emergency obstetric and newborn care facilities (CEmONC) when an escalation of care is needed for blood transfusion and or cesarean section [19,47,52]. A notable strength of this study is the applied cross-country learnings.”

---

## [Decision Letter · Decision Letter 1]

19 May 2026

Cross-context qualitative synthesis of a mobile obstetric referral emergency system (MORES) in Ghana and Liberia

PONE-D-25-52452R1

Dear Dr. Bjelland,

We’re pleased to inform you that your manuscript has been judged scientifically suitable for publication and will be formally accepted for publication once it meets all outstanding technical requirements.

An invoice will be generated when your article is formally accepted. Please note, if your institution has a publishing partnership with PLOS and your article meets the relevant criteria, all or part of your publication costs will be covered. Please make sure your user information is up-to-date by logging into Editorial Manager at Editorial Manager® and clicking the ‘Update My Information’ link at the top of the page. For questions related to billing, please contact billing support.

Kind regards,

Maurine Rofhiwa Musie, PhD

Academic Editor

PLOS One

Additional Editor Comments (optional):

The authors have addressed the comments of the two reviewers and the Academic Editor satisfactorily. The revisions have significantly improved the quality, clarity, and scholarly contribution of the manuscript. The concerns raised during the review process have been adequately resolved, and the manuscript now meets the required standards for publication. I therefore recommend acceptance of the manuscript for publication.

Reviewers’ comments:

Reviewer’s Responses to Questions

**Comments to the Author**

1. If the authors have adequately addressed your comments raised in a previous round of review and you feel that this manuscript is now acceptable for publication, you may indicate that here to bypass the “Comments to the Author” section, enter your conflict of interest statement in the “Confidential to Editor” section, and submit your "Accept" recommendation.

Reviewer #1: All comments have been addressed

Reviewer #2: All comments have been addressed

2. Is the manuscript technically sound, and do the data support the conclusions?

Reviewer #1: Yes

Reviewer #2: Yes

3. Has the statistical analysis been performed appropriately and rigorously? 

Reviewer #1: N/A

Reviewer #2: N/A

4. Have the authors made all data underlying the findings in their manuscript fully available?

Reviewer #1: No

Reviewer #2: Yes

5. Is the manuscript presented in an intelligible fashion and written in standard English?

Reviewer #1: Yes

Reviewer #2: Yes

6. Review Comments to the Author

Reviewer #1: The authors have responded comprehensively and constructively to the reviewers’ comments. The revisions have significantly improved the clarity, methodological transparency and overall framing of the manuscript. Overall, the manuscript is now clearer, more methodologically sound and aligned with its stated aims. I have no further substantive concerns and recommend the manuscript for acceptance. No concerns regarding research ethics, dual publication, or publication ethics are noted.

Reviewer #2: The points initially raised have been satisfactorily addressed.

The study limitations are better framed and stated with further clarity.

7. PLOS authors have the option to publish the peer review history of their article (what does this mean?). If published, this will include your full peer review and any attached files.

Reviewer #1: **Yes:**Zimkhitha Final Juqu

Reviewer #2: No

---

## [Editor Report · Acceptance letter]

PONE-D-25-52452R1

PLOS One

Dear Dr. Bjelland,

I’m pleased to inform you that your manuscript has been deemed suitable for publication in PLOS One. Congratulations! Your manuscript is now being handed over to our production team.

Lastly, if your institution or institutions have a press office, please let them know about your upcoming paper now to help maximize its impact. If they’ll be preparing press materials, please inform our press team within the next 48 hours. Your manuscript will remain under strict press embargo until 2 pm Eastern Time on the date of publication. For more information, please contact onepress@plos.org.

Kind regards,

on behalf of

Dr. Maurine Rofhiwa Musie

Academic Editor

PLOS One